# Adolescent Behaviours and Their Relationship to the Risk of Developing Eating Disorders

**DOI:** 10.3390/healthcare11040624

**Published:** 2023-02-20

**Authors:** Joaquín Tarifa Pérez, Montserrat Monserrat Hernández, Ángeles Arjona Garrido, Darío Salguero García, Juan Carlos Checa Olmos

**Affiliations:** 1Junta de Andalucía, 18071 Granada, Spain; 2Laboratory of Social and Cultural Anthropology, University of Almeria, La Cañada, 04120 Almería, Spain

**Keywords:** eating disorders, adolescents, SCOFF index, social network, family

## Abstract

Eating disorders (ED) in adolescents represent a significant problem in today’s society, with multiple factors influencing them, such as predisposing factors, precipitating factors and perpetuating factors. Objective: This paper aimed to determine the relationships between some of the factors considered to be predisposing and precipitating in terms of the development of ED in adolescents and to relate them to the SCOFF index. Participants: The sample was made up of 264 subjects aged between 15 and 19 (48.8% females and 51.1% males). Methods: This study was conducted in two phases. In the first study phase, a descriptive analysis of the sample was performed, encompassing the frequencies of the independent variables and dependent variable (ED). In the second phase of study, we created several linear regression models. Results: A total of 11.7% of adolescents are at high risk for ED, and the variables that predict the variability of manifesting the danger of ED are the following: physical self-conception and family relationships. Conclusions: This work shows the need to approach eating disorders in a multidisciplinary way (biological and social), since this will enable the disease to be better conceptualized and prevention guidelines to be more effective.

## 1. Introduction

Eating disorders (ED) in adolescents are a significant problem in today’s society [1,2,3]. The peak incidence is between 15 and 25 years of age [4,5] and mostly in females [4,6]. Current explanations argue that women are under greater pressure with regard to physique and the standards of beauty in society [7], although this trend is changing and similar pressures regarding the male appearance are becoming more prevalent [8].

There are multifactorial influences on ED [9,10]: (1) predisposing factors (genetics, age, socio-cultural factors, family and friends, etc.); (2) precipitating factors (physical changes, stress processes, etc.); and (3) perpetuating factors (apathy, depression, introversion, sport, etc.).

The period of adolescence often coincides with an identity crisis that leads to fears of new situations and anxiety if adolescents feel they do not fit within societal standards, for example, of beauty. The search for social identity—how I identify myself in opposition to others and how others identify me (from physical to intellectual aspects) [11,12]—becomes important in both face-to-face and virtual interpersonal relationships through the use of social networks (RRSS) [13,14]. Recent studies have shown a significant relationship between the frequency of use of RRSS and the risk of developing EDs [13,14,15].

Social networks are designed in an open way where people can act as simple viewers or generators of content, which are mainly images. These networks have contributed to the consolidation of the idea that in order to be socially successful, the physical aspect of an individual is fundamental and must meet the societal standard of “slim”. Hence, falling outside this ideal model of beauty often has a negative influence on an adolescent’s perception of their body image [16,17,18]. In other words, the first step towards an ED is dissatisfaction with one’s physical self-image, which leads the adolescent to modify their eating pattern [19,20] and, on online networks, to use filters and/or retouching to modify their physical appearance. In both cases, the aim is to portray a supposedly perfect image that pleases as many people as possible [21].

Many of the other related factors are the result of endogeneity, where family relationships are of great importance [22]. Early experiences, identification processes with parental figures, the way in which adults help to resolve developmental conflicts, etc., become important. Thus, a healthy and functioning family can help its members to cope with illnesses and problems, in the present and in the future [23,24].

The circumplex model of marital and family systems [25] is followed in the current study to measure the factors of cohesion (time sharing, interests and recreation); flexibility (rigidity in the family system); and communication (empathy and capacity for dialogue). In this case, the relationship with the family has been studied and measured regarding its role in the manifestation of ED [26].

Nevertheless, despite the numerous studies that focus on the dangers of EDs, as with most mental conditions, their diagnosis is clinical and must be made by an expert. Even so, we note two aspects worth highlighting regarding prevention. On the one hand, these tests provide information and are carried out when the subject is already manifesting the illness (even if it is in its early stages), so prevention is left in the hands of the social environment in which the adolescent finds themselves (school, family, peer groups, etc.). On the other hand, the questionnaires used for diagnosis are considered long and, when short surveys are used, such as the SCOFF index [27], they focus on dimensions related to physical appearance or food.

We therefore consider the early detection of EDs to be difficult due to the multifactorial implications. However, the aim of this study is to determine the relationships between some of the predisposing and precipitating factors in the development of EDs in adolescents aged 15–16 years [2,3], such as gender, relationship with family, use of social networks, self-conception and social image, and to relate them to the SCOFF index, validated as an instrument for detecting the risk of developing EDs.

## 2. Method and Data

### 2.1. Participants

The population of our study (young people from Almeria aged between 15 and 19 years old) was 33,133 [28]. An error of ±5 (90% confidence level) was assumed, resulting in 264 participants (48.8% female and 51.1% male). Additionally, it should be noted that 91% of these were of Spanish nationality.

### 2.2. Instrument and Procedure

The self-reporting survey completed by the participants consisted of several sections that were designed to collect the data for this study. Sections included socio-demographics, the use of social networks, the type of content followed on social networks, physical self-conception, social image, family relationships and the SCOFF index [27,28,29,30].

Although some authors [30] suggest using Likert-type graduated response scales, other researchers [31] find that a dichotomous format provides almost as much information as a polytomous format. Furthermore, when working with adolescents, reduction to dichotomous categorical scales may optimise measurement because it better accounts for response variability [29]. Moreover, given that the aim is to establish relationships between the new variables and the results obtained in the SCOFF index (also embedded in the questionnaire), we considered it appropriate to establish similar measurement parameters.

The items were drafted and selected on the basis of information obtained from a panel of experts and a review of the literature. Subsequently, the group of authors of this study, with experience in EDs and adolescents, consisting of a psychologist, a secondary school teacher, two sociologists and a nutritionist, corrected defects detected in terms of the relevance of the items, items needed for each model, the accuracy of the questions and terminology used.

As a dependent variable, we used the score obtained in the SCOFF questionnaire. This is a five-item self-administered questionnaire with dichotomous answers. Each affirmative answer is valued at 1 point, and the proposed cut-off point for screening in the general population is equal to or higher than 2 points. The questions are as follows: (1) do you make yourself throw up because you feel lighter; (2) do you worry about losing control over the amount of food you eat; (3) have you lost more than 7 kg in three months or less; (4) do you think you are fat; and (5) would you say that food dominates your life? For example, thinking about eating constantly.

Twelve questions divided into five sections were used as independent variables referring to dimensions related to the activities and behaviours of adolescents, comprising dichotomous, true/false and yes/no questions.

The use of social networks was divided into two sections (type of use and type of content followed), and, as was shown in the introduction section, there may be a difference between the two concepts.

The type of use was assessed with the following questions: (1) I use social networks mainly to upload photos of myself; (2) I use social networks to report on my life; and (3) I use social networks mainly to see photos or videos of other people.

The type of content viewed was assessed by the following: (1) celebrities; (2) content creators followed for reasons other than their physical traits; and (3) beautiful and/or good-looking people.

Physical self-conception was assessed with the following statements: (1) I would like to be thinner; (2) I would like to look like someone else in my class or school; and (3) I would like my physique to be different.

Regarding the assessment of social image, we took into account questions that did not only encompass the physical, as is verified in the attached bibliography [11,15,21], and value comparisons can be established as related to other concepts, through statements such as the following: (1) I think that others respect me; (2) others accept me as I am; and (3) I like what others think about me.

Regarding family relationships, following the recommendations of Olson and collaborators and their approach of the circumplex model of marital and family systems [24], we established three questions that were related to the three dimensions described: (1) cohesion (we usually undertake activities as a family); (2) flexibility (at home I am understood and supported); and (3) communication (at home I talk to my parents when I need to).

Subsequent to the development of the questionnaire, content validity was analysed by two external reviewers and a Cohen’s kappa value of 1 (sig < 0.001) was obtained. Both agreed that the items “I would like to be like someone else” and “At home I talk to my parents” were poorly worded and were changed to “I would like to be like someone else in class or at school” and “At home I talk to my parents when I need to” (see Table 1).

The information was obtained randomly. Between September and October 2022, a questionnaire was administered to a series of fourth-year secondary school classes from eight different schools in the province.

The questionnaire was administered using LimeSurvey (https://www.limesurvey.org) (accessed on 1 September 2022). The participants were informed of the purpose of the research, the voluntary nature of their participation, the mechanisms that guarantee their anonymity and data protection laws. In addition, as they were minors, the legal guardians of each student signed a prior consent form.

### 2.3. Statistical Analysis

The data were analysed in the SPSS-27 statistical analysis package. Initially, a descriptive and frequency analysis was carried out to observe the results on a generalised scale in order to examine the means obtained for each of the proposed questions. Subsequently, a linear regression analysis was performed to investigate the relationship between the different questions posed and the results of the SCOFF index. The former were divided and classified into the following categories: use of social networks, type of content followed on social networks, self-conception, social image and family relationships.

## 3. Results

Table 2 shows the descriptive characteristics of the sample in terms of the responses to the questionnaire, divided by sections/blogs. With respect to the variables related to the use of social networks, close percentages can be observed between male and female adolescents with respect to uploading photos of themselves (22.5% female and 19.2% male) and viewing photos or videos of other people (64.3% female and 58.5% male). However, there is a clear difference when it comes to reporting on their life through social networks, with female adolescents reporting higher percentages than their male counterparts (82.2% and 21.5%, respectively).

In terms of the content viewed on social networks, 52.2% of respondents followed a person because they are famous, 44.7% followed a person because they are good looking and have a good physique, and 44.7% followed people for reasons other than their physical appearance. As the answers to these questions were not mutually exclusive, we can see that there are a variety of motivations for these young people to follow someone on a social network. In terms of gender, we observe similar percentages in all the questions, with the biggest difference in the category of following someone because they are famous, which was the case for 58.5% male and 45.7% female adolescents.

When we refer to physical self-conception, we see that 54.2% of those surveyed do not like their body image and would like to change it (55.8% female and 52.6% male). We also observe that almost half of those surveyed (46.6%) would like to be thinner, and there was not much difference between female and male adolescents in this category (50.4% and 43%, respectively). This contrasts with the 24.4% who want to look like someone else in their class or school.

When we discuss social image, we go beyond the physical. Here, we attempt to analyse how young people think they are seen by their peers. In this respect, 59% feel accepted (63.6% female and 54.8% male adolescents) and 64.4% like what others think of them (with no notable differences between male and female adolescents). However, only 29.2% of respondents believe that they are respected (32.5% female and 26% male).

Regarding family relationships and following the guidelines of the circumplex model of marital and family systems [24], we established three questions related to the following: (1) family cohesion, where 37.5% of respondents state that they undertake activities as a family (34.1% female and 40.7% male); (2) flexibility or capacity for understanding on the part of guardians, where 66.6% of the young respondents state that they feel understood and supported (66% female and 67.4% male); and (3) communication, where 59.5% of the participants state that they feel that they can talk to their parents/guardians when they need to (56.6% female and 62.2% male).

With regard to the SCOFF index, after its administration in our experimental population, 31 adolescents had scores higher than 2 (considered at risk of developing EDs). This corresponds to a prevalence of 11.74%. Of these, 18 were female (13.9% of all females) and 13 were male (9.6% of all males).

### Relationship between the Results Obtained in the SCOFF Questionnaire and the Questions Observed

In response to the objective of uncovering the possible variables that predict the manifestation of an ED, we created several linear regression models. As can be seen in Table 3 and Table 4, all models are significant. Table 4 shows that each model is made up of the related questions added to the previous section. 

Gender was introduced as a control variable, as can be seen. Although the data show that being a woman is a recognised factor, it is not significant.

In Model 1, variables related to the use of social networks are introduced. We observe that the categories of uploading photos and not uploading information about themselves show significant relationships (*p* > 0.05) with high results in the SCOFF index.

In Model 2, we introduce the variables related to the type of profile the adolescents follow on social networks. In this case, none of the three linked responses obtained significant results. However, the variables “reporting about my life” (negative) and “uploading photos of myself” remain significant (*p* > 0.05).

In Model 3, we include self-conception. Here, we observe that wanting to be thinner and not using social networks to report on “my life” are highly correlated (*p* > 0.005) with high scores on the SCOFF scale.

In Model 4, the social image dimension is added and no significant changes are obtained with respect to the previous models. There is a high relationship of the use of social networks to report on “my life” (negative) and wanting to be thinner with SCOFF scores.

When we add in family relationships (Model 5), we observe that, with very high significance, not being able to talk to parents when needed is related to high SCOFF scores.

## 4. Discussion

### 4.1. Main Findings

The surveyed population reflects the proportions of the Spanish population in terms of the risk of suffering from EDs (specifically anorexia nervosa and bulimia). The fact that 11.7% of the population analysed (18 female and 13 male adolescents) obtained high scores highlights a reality that has been widely publicised by professionals [1,2,3]. There is a need to prevent the onset of these disorders. Moreover, as recent studies have shown [4,5], this problem is not only confined to women; an increasing number of men are at risk of developing EDs.

In this regard, in our research, we wanted to analyse the degree of influence that different dimensions related to young people may have on the risk of developing ED.

There are currently publications that relate the use of social networks and the type of content consumed, which is linked to physical activity, with the risk of developing an ED [14,15,16,31], but in our study, these relationships were not significant.

We understand that this situation may be due to several factors, even if they are contradictory: first, the educational system and the family environment are having positive effects on young people when it comes to breaking with the standards of beauty dictated by the media. Secondly, the homogenization of photos of sculpted bodies manipulated using filters or other computer programs by content creators is creating weariness and disbelief among young consumers. In any case, these or other explanations for the lack of a significant relationship between the above-mentioned factors require a more in-depth study, as well as a longitudinal analysis.

We did observe strong links in terms of the content uploaded (I do not report on my life, and I upload pictures of myself). These data may be interesting if we analyse how photos are posted today with the use of filters to disguise features and/or imperfections and the prevalence of poses that seek to highlight thinness and possibly manifest the desirable physique of these content creators. In this case, these results can be related to those found in the “self-conception” section, where wanting to be thinner is highly related to the risk of suffering from an ED.

However, when we analyse the descriptive data obtained, we realise that although a high percentage of these students do not accept themselves physically, not as many use classmates or schoolmates as references. These results suggest that ideals are not usually found in the circle of peers and that they can be sought in famous people or others who can be reached through new technologies and social networks [16,17,31].

If we go beyond the physical aspect, we can see that social image (which influences self-esteem and self-conception) does not influence the risk of developing EDs. However, analysing the data in terms of percentages, we can see that almost half of those surveyed do not feel accepted or loved as they are, and furthermore, only 29% feel that they are respected. These values coincide with recent studies that examine adolescent problems [11,12,13], although these studies do show significant relationships regarding the manifestation of EDs.

When we come to the section concerning family relationships in our study, we observe that the most important factor in the manifestation of an ED is not being able to talk to parents or guardians when needed. Current studies show that a bad relationship or even a lack of affection can be potential contributors to the development of EDs [5,25,26].

In conclusion, we would like to point out that, although there is extensive literature on this subject, much of it tends to focus on the factors influencing the manifestation of EDs separately (family, internet, peers, etc.). Following the models of Steiger et al. and Leung et al., published in 1996, which advocated enrichment by including new factors, we have sought to update these factors to the current circumstances that surround young people and thus propose a more complete analysis.

### 4.2. Limitations

Although this study shows the influence of certain predictor variables on the risk of developing EDs and thus provides professionals with material to work on prevention, this study has some limitations.

First, when measuring the risk of developing EDs, the SCOFF index was used, which shows the risk of developing anorexia nervosa or bulimia nervosa. There is no score to detect people affected by binge eating disorders. In this case, there may be a higher percentage of respondents who are at risk of developing a binge eating disorder, which has not been taken into account.

Secondly, this study did not follow up on the type of consumption by students on social media. Perhaps an additional qualitative study, consisting of focus groups and/or interviews, could generate more information to contrast with what was observed.

Thirdly, further research with a larger study population is needed to confirm the results obtained.

## 5. Conclusions and Implications

Previous research has revealed the importance of young people’s environment as well as the use of social networks in the development of an ED. This study shows that although there are social pressures and Internet pressures, self-conception, self-esteem and family relationships become strong weapons in the fight against this growing pandemic among young people. For all of these reasons, we call on the professionals who work with adolescents to make it a priority to establish spaces for dialogue and connection as well as to include techniques for the promotion of self-esteem and the improvement of self-conception in the teaching curriculum in a cross-cutting manner.

On the other hand, we want to emphasize that these professionals cannot achieve this alone, and their work must be a continuation of what has been created in the family home; from childhood, it is necessary to promote self-esteem (in relation to parts of our body that we like and those that we do not), accept mistakes and failures as something natural, take care of the self, always set realistic and achievable goals, work on frustration, recognise our own merits and worth within society, promote a positive body image, accept our differences, etc.

## Figures and Tables

**Table 1 healthcare-11-00624-t001:** Content validity analysis (results of external reviewers).

	Value	Asymptotic Standard Error ^a^	T Approximate ^b^	Approximate Significance
Kappa measure of agreement	1.000	0.000	5.745	0.000
Valid cases	15 items			

^a^ = note that SPSS reports the wrong standar error; ^b^ = about the independents variables.

**Table 2 healthcare-11-00624-t002:** Descriptive characteristics of the sample. Mean (SD) and differentiation by sex (N and %).

Characteristics	Mean (SD)	Sample (N)	Sample (% of Total for Each Sex)
Sex		Female (129) Male (135) Total (264)	
Use of social networks			
I use social networks mainly to upload photos of myself	M = 0.2 (0.4)	Female (29) Male (26) Total (55)	Female (22.5) Male (19.2) Total (20.8)
I use social networks to inform about my life	M = 0.2 (0.4)	Female (106) Male (29) Total (135)	Female (82.2) Male (21.5) Total (51.1)
I use social networks mainly to see photos or videos of other people	M = 0.6 (0.5)	Female (83) Male (79) Total (162)	Female (64.3) Male (58.5) Total (61.4)
Type of content I follow on social networks			
I follow someone on social networks if they are famous	M = 0.5 (0.5)	Female (59) Male (79) Total (138)	Female (45.7) Male (58.5) Total (52.3)
I follow someone if they provide content beyond physical traits	M = 0.4 (0.49)	Female (59) Male (59) Total (118)	Female (45.7) Male (43.7) Total (44.7)
I follow someone on social networks if he/she is good looking and has a good physique	M = 0.6 (0.5)	Female (79) Male (83) Total (162)	Female (61.2) Male (61.5) Total (44.7)
Physical self-conception			
I would like to be thinner	M = 0.4 (0.4)	Female (65) Male (58) Total (123)	Female (50.4) Male (43) Total (46.6)
I would like to look like someone else in my class or school	M = 0.2 (0.4)	Female (32) Male (32) Total (64)	Female (24.8) Male (23.7) Total (24.2)
I would like my physique to be different	M = 0.5 (0.4)	Female (72) Male (71) Total (143)	Female (55.8) Male (52.59) Total (54.2)
Social image			
I believe that others respect me	M = 0.3 (0.4)	Female (42) Male (35) Total (77)	Female (32.5) Male (26) Total (29.2)
Others accept me as I am	M = 0.5 (0.5)	Female (82) Male (74) Total (156)	Female (63.6) MAle (54.8) Total (59.1)
I like what others think about me	M = 0.6 (0.5)	Female (86) Male (84) Total (170)	Female (66.6) Male (62.2) Total (64.4)
Family relationships			
At home we undertake a lot of activities together	M = 0.4 (0.5)	Female (44) Male (55) Total (99)	Female (34.1) Male (40.7) Total (37.5)
At home I talk to my parents when I need to	M = 0.6 (0.5)	Female (73) Male (84) Total (157)	Female (56.6) Male (62.2) Total (59.5)
At home I am understood and supported	M = 0.6 (0.5)	Female (85) Male (91) Total (176)	Female (66) Male (67.4) Total (66.6)
SCOFF index (>2 points)	M = 1.04 (1.2)	Female (18) Male (13) Total (31)	Female (13.9) Male (9.6) Total (11.7)

Abbreviation: SCOFF—Sick, Control, One stone, Fat, Food.

**Table 3 healthcare-11-00624-t003:** Summary of the general model.

Model	R	Sig	Durbin–Watson
1		0.043	
2		0.049	
3		<0.001	
4		<0.001	
5	0.482	<0.001	1.829

**Table 4 healthcare-11-00624-t004:** Explanatory models of the degree of risk of developing EDs.

Variables			Model 1	Model 2	Model 3	Model 4	Model 5
Individuals	Sex (male)	−0.17	−0.42	−0.24	0.59	0.039	0.071
RRSS use	To upload photos of myself		0.319 *	0.334 *	0.272 *	0.289 *	0.261 *
	To report on my life		−0.385 *	−0.374 *	−0.421 ***	−0.423 **	−0.499 ***
	To see other people’s photos		−0.106	−0.094	−0.114	−0.099	−0.070
Who I follow on RRSS	Famous people			−0.142	−0.220	−0.205	−0.152
	Creator who provides content beyond physical traits			0.084	0.043	−0.099	−0.019
	Good looking and with a good physique			−0.142	−0.220	−0.205	−0.0152
Self-conception	I would like to be slimmer				0.794 ***	0.800 ***	0.752 ***
	I would like to look like someone else in my class or school				0.166	0.178	0.144
	I would like my physique to be different				0.218	0.180	0.261
Social image	I believe that others respect me					−0.097	−0.016
	I believe that others accept me as I am					−0.113	−0.146
	I like what others think about me					−0.004	0.174
Family relationships	At home we undertake a lot of activities together						−0.172
	At home I talk to my parents when I need to						−0.575 ***
	At home I am understood and supported						−0.127

Abbreviations: RRSS—social networks. * *p* < 0.05; ** *p* < 0.01; *** *p* < 0.005.

## Data Availability

Not applicable.

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
