# Peer review of "Adolescent Behaviours and Their Relationship to the Risk of Developing Eating Disorders"

_healthcare, 2023, doi:10.3390/healthcare11040624_

Round 1

Reviewer 1 Report

The authors present data relating to an excellent starting point in relation to the difficult world of eating disorders. The research is well structured, the sample examined is particularly large and therefore indicative for the conclusions drawn.

I would recommend expanding the introductory section by further examining all those aspects predisposing to the development of ED. In particular, the factors that lead adolescents to not see themselves adequate with respect to the ideals proposed by the world around them. This is to make the introduction more complete for the reader.

I also advise authors to have the manuscript proofread by an English native to improve the language

Author Response

The topic of social factors in the development of ED was expanded a little. However, it was not extensive since referee 2 suggests making a more synthetic introduction.

We use the MDPI English service to improve the language.

Reviewer 2 Report

Dear authors,

Congratulations for your interesting research !

The Introduction section can be rearranged and be presented more synthetic. Your approach is quite challenging and must be valued more in the discussion part. Conclusions are briefly presented, and I think it will be clearer for readers to emphasize on your own perspective about the relationship between eating disorders and different types of teenagers' habits after completing this research !

Author Response

The introduction has been synthesized a bit.

Conclusions are briefly presented, and I think it will be clearer for readers to emphasize on your own perspective about the relationship between eating disorders and different types of teenagers' habits after completing this research.

Done.

Reviewer 3 Report

Thanks for the opportunity of reviewing the present manuscript. 

The authors aimed to determine the possible relationships between factors predisposing to ED and relate them to the SCOFF index.

The topic is relevant and the idea promising, however the manuscript may require some minor revision before being fully considered for publication.

Abstract:

The structure of the abstract is confusing, especially in the second part (from line 14 to 18). The authors should consider rephrasing, make the sentences more understandable and follow a logic information flow bringing the reader from the aim to the conclusion.

Introduction:

Sentence line 29-35 is too long and is easy to lose the sense of the paragraph. Consider to rephrase it and divide in two separate sentences.

Methods

the authors may consider to find an alternative to the subtitle 2.3 (line 126)

Discussion

The authors may consider to extend their discussion including possible and additional hypotheses explaining their results (e.g., why the relationships between SN and ED resulted not strong? what can be the reason behind this? and which other emerging factor may be considered in future studies?)

Author Response

Improved abstract.

Introduction:

Sentence line 29-35 is too long and is easy to lose the sense of the paragraph. Consider to rephrase it and divide in two separate sentences.

Separated. And separated and modified to comply with referee 1's request.

Methods

-the authors may consider to find an alternative to the subtitle 2.3 (line 126)

Subtitle 2.3. has been eliminated, it was a typo. Renumbered subtitle 2.4.

Discussion

The authors may consider to extend their discussion including possible and additional hypotheses explaining their results (e.g., why the relationships between SN and ED resulted not strong? what can be the reason behind this? and which other emerging factor may be considered in future studies?)

An explanation has been added.

Reviewer 4 Report

Dear      Authors,

the work presented to me for evaluation is very interesting and meets the thematic requirements of the journal. The work is prepared very carefully, but some technical errors need to be corrected. Please review the paper carefully and comply with the requirements of the journal, especially:

       -correct the references as required by the journal

-          - all abbreviations in the tables should be explained

        -in the RESULTS: table caption is not needed

-              - all abbreviations used for the first time must be explained (line 193)

-          - in tables: please replace the commas with full stops where necessary

These are minor errors and after making corrections, the work should be published.

Author Response

the work presented to me for evaluation is very interesting and meets the thematic requirements of the journal. The work is prepared very carefully, but some technical errors need to be corrected. Please review the paper carefully and comply with the requirements of the journal, especially:

       -correct the references as required by the journal

Checked bibliographic references

-          - all abbreviations in the tables should be explained

Abbreviations introduced in the table captions

        -in the RESULTS: table caption is not needed

Removed table footers, except in table 4: *P; **P

-              - all abbreviations used for the first time must be explained (line 193)

The abbreviation has been eliminated and the correct abbreviation has been introduced (ED).

Previously your acronym was clarified

- in tables: please replace the commas with full stops where necessary

- changed